# A Security Management and Control Solution of Smart Park Based on Sensor Networks

**DOI:** 10.3390/s21206815

**Published:** 2021-10-13

**Authors:** Yue Zhao, Bo Tian, Yiru Niu, Hao Zhang, Zhongqiang Yi, Ruiqi Zeng

**Affiliations:** 1Science and Technology on Communication Security Laboratory, Chengdu 610041, China; tb_30wish@163.com (B.T.); niu_yiru@outlook.com (Y.N.); 13219067337@163.com (H.Z.); zengruiqi@sina.com (R.Z.); 2No.30 Research Institute of China Electronics Technology Group Corporation, Chengdu 610041, China; yzq0003@126.com

**Keywords:** sensor networks, smart park, security management and control, joint authorization and dynamic access control, Merkle tree

## Abstract

As a typical application of sensor networks, there exist many information security problems in smart parks, such as confusion of personnel access, lack of security management, disorderly data flow, insufficient collection of audit evidence, and so on. Aiming at the scenario of personnel and equipment moving in different areas of smart parks, the paper proposes a joint authorization and dynamic access control mechanism, which can provide unified identity management services, access control services, and policy management services, and effectively solve the problem of multi-authorization in user identity and authority management. The license negotiation interaction protocol is designed to prevent common network attack threats in the process of identity authentication and authority management. In order to realize the tamper-proof storage of personnel and equipment movement trajectory, the paper also designs a movement trajectory traceability protocol based on a Merkle tree, which solves the problems of internal personnel malicious attack, trusted third-party dependency bottleneck, high overheads of tracking algorithms, and so on. The experimental results show that compared with the current security control mechanisms for sensor networks, the joint authorization, and dynamic access control mechanism can support multi-party authorization and traceability, while the overhead it generates in initialization, encryption, decryption, and key generation steps are basically the same as other mechanisms do.

## 1. Introduction

Sensor networks have been fully used in smart parks to enable the integration of physical facilities such as entrance guards, gates, and cameras with smart park information systems. Smart parks rely on sensor networks to manage and control people, equipment, and infrastructure in the park in a more granular and dynamic manner in real time [1]. However, as an open place, smart parks suffer from chaotic personnel access and a lack of security management. Moreover, the sensor network collects, transmits, and aggregates a large amount of multi-source heterogeneous data in the park [2], which also suffers from disorderly flow and a lack of audit forensics. As the application services carried by smart parks become more complex, these problems bring more risks, which seriously hinder the development of smart parks [3].

In this work, we focus on the important role that information security technology can play in smart parks based on sensor networks. We aim to accomplish two main goals: to realize fine control of personnel movement before users enter a certain area of the park, and to form tamper-proof and forgery-resistant log records of personnel and equipment movement trajectory in the park. Therefore, we propose a joint authorization and dynamic access control mechanism (JADA) for the scenario in which people and devices move in different areas of smart parks. Joint authorization means there are different administrators who can restrict the access rights of users so that several policies can work together to determine whether a user can access a certain area or not, which will lead to a problem of policy conflict. The joint authorization model should be able to accurately describe the permissions and related constraints that a subject is given to access a specific object. Meanwhile, secure and effective technical measures should be taken to ensure that the above authorization and constraints can be properly enforced. Dynamic access is mainly manifested in the three following aspects.

(1)The dynamic of permission acquisition.

The subject’s permission is dynamically derived in the process of its access to the park based on specific policies.

(2)The dynamics of the basis for authorization decisions.

Unlike the identity-based access control (IBAC) mechanism, there are many variable factors that decide, on basis of the data collected by the sensor networks, whether to authorize or not.

(3)The changeability of authorization.

In the process of the visitor’s access to different areas of a smart park, it may be necessary to change the authority possessed by the visitor or even suspend the authorization.

The paper is organized as follows: Section 2 overviews the related work; Section 3 describes the system model; Section 4 presents the refined authority control mechanism before personnel movement in smart parks, and proposes the clarification mechanism for the whole process of activity track after personnel movement in smart parks; Section 5 describes the experiment environment, and carries out performance analysis; Section 6 concludes the paper.

## 2. Related Works

In recent years, the information security problems that exist in smart parks have received increasing attention. Gope et al. proposed a radio frequency identification (RFID) lightweight authentication scheme for sensor networks in a smart park environment, which required less time for the authentication process and could effectively resist replay attacks, forgery attacks, DDoS attacks, and location tracking attacks [4]. 

Wang introduced the current authentication protocols commonly used by the three parties of sensor network end-devices, edge-devices, and control center, and proposed a key negotiation scheme based on Shamir’s secret sharing scheme to design the key negotiation scheme among the above three parties; also giving an idea about a group signature-based access authentication protocol that is designed to protect the privacy of end devices [5]. 

Barni et al. investigated how to combine signal processing and cryptographic primitives, with a focus on the application of secure multi-party computation in the access control model to protect the biometric data privacy of users [6].

Zhao et al. presented an innovative approach to evaluate the security of sensor networks based on the call chain technique test, which supports the automatic construction of call chains in terms of type, temporal order and causality, and implements the detection of potential security risks in sensor networks according to different test paths [7]. 

Liu et al. proposed the novel channel-hopping pattern for sensor networks in smart cities as a physical-layer security technique that can be integrated with upper-layer security policies such as encryption and authentication [8].

However, the above research all failed to effectively solve the security control problems in smart parks. First, there is a lack of unified permission control technology and a dynamic and efficient authorization and revocation capability, and with the exponential growth in the number of users, there arises disorderly management of personnel and equipment in the park and even paralysis of the information system of the smart park if the permission of illegal users or devices cannot be revoked in a timely manner. Second, the current centralized data processing models used in smart parks are inefficient. The detection and feedback cannot meet the requirements of timeliness, which leads to the low effectiveness of smart parks' operation and maintenance management [9]. More data processing equipment and resources are often required to achieve efficient and low latency processing effects, which also becomes a financial burden when building a smart park.

On the other hand, in the event of public safety incidents in smart parks, most of the existing security control schemes for smart parks do not consider the clarification of the mobile trajectories of people and equipment. Even though some literature has considered using tracking algorithms or introducing trusted third parties to achieve forensics and traceability [10,11], tracking algorithms have a large computational overhead for sensor networks, while introducing a trusted third party sometimes becomes a new security bottleneck in practical applications.

Therefore, we considered using a Merkle tree to trace the movement process of personnel in the smart park. Compared to the traditional system log forensic technology [12], the specific user behavior can be quickly located using the method of Merkle tree when an abnormal situation occurs, providing strong support for tracking and tracing. Merkle trees are rarely used for log storage and are typically used for operational log storage of cloud data. Mao et al. [13] proposed a position-aware Merkle tree that can compute the root node without retrieving the whole Merkle tree; in this way, the storage complexity, computation cost, and communication cost will be reduced. Yang et al. [14] propose a multi-grained log auditing scheme to improve the efficiency, which introduces a Merkle Hash Tree structure to support multiple data granularity in logs. Based on the multi-grained log, they present a multi-grained data confidentiality auditing. But there has been no example of using a Merkle tree for tracking the behavior of users in the application systems of smart parks.

A novel classification technology of gender and age from images was proposed in [15]. We focus on the fine control of users and tracing the movement trajectory of personnel and equipment after we recognize the user’s identity. The image acquisition and image classification technologies have given us some inspiration, and we can use the system proposed, which can classify gender and age, to enrich the function of our face recognition module.

In conclusion, the existing schemes cannot support dynamic authorization, minimal authorization, and joint authorization. They cannot solve the problem of conflict policy resolution after joint authorization. Still, JADA proposed in this paper can provide efficient traceability and incident recovery for park personnel, and solve the problems of malicious insider attacks, trusted third-party dependency bottlenecks, and high overhead of the tracking algorithm by introducing an immutable record mechanism.

## 3. System Model

The sensor network-based security control model of smart parks and its processes are shown in Figure 1. Sensor devices such as cameras and temperature transducers are always in operation, and the intelligent gateway (also known as sensor network gateway) is used to intercept the access requests submitted by the user subject and is responsible for verifying the subject’s identity information. The intelligent gateway is the interaction interface between the subject and the license server [16,17]. The intelligent gateway does not have the user’s identity and policy information and needs to submit a request to the license server via sensor networks to negotiate an available license. The license is both a key element to achieve authorization and an effective implementation and carrier of the policy, serving as a permission contract between the user and the park administrators. The policy enforcement point (PEP) is the program component of policy usage control [18], responsible for generating authorization requests and executing authorization responses returned by the policy decision point (PDP). The PDP is responsible for executing policy judgments and decisions when the user subject’s usage request is allowed. In order to provide more accurate decisions, the PDP may also need to query the policy information point (PIP) to collect descriptive information about attributes or to find policy information provided by other third parties. A policy administration point (PAP) is a system entity that makes and manages policies and policy sets [19].

The attribute-based access control (ABAC) mechanism supports fine-grained access control on whether a user can access a certain area of the smart parks. The attributes mainly include subject attributes, object attributes, environment attributes, and operation attributes [20,21]. In the smart park scenarios, the subject refers to users accessing the park; the object refers to the access control device of the accessed park area; the operation refers to whether the user has the right to enter the park and the access control device is turned on or off; and the environment refers to the current time, date, or the actual operating state of the park when access occurs. The policy refers to the set of security policies that the park administrator follows to implement a certain operation on the subject (i.e., park visitor), and mainly contains two decision factors: condition and authorization.

The first half of a policy statement is a condition, which can generally be expressed by a judgmental expression, and the second half of a policy statement is a permission, which can have the operation permission of the second half when and only when the condition expression is true [22]. A policy statement generally consists of an authorization predicate expression and a conditional function expression [23]. Conditional expressions are often based on the properties of the subject, object and environment. For example, the following policy statement can be defined as,

IF s.department = “computer department” AND s.role “teacher”

THEN enter Room 101

This policy allows subjects who conform to the “computer department” and whose role is “teacher” to enter room 101. Such a policy can assign one or more permissions to a group of users with partially the same attributes. Changes in user attributes will affect the permissions they can have. In addition to the strategy composed of authorization predicates and conditional expressions, predicate functions that update attributes should also be defined due to the variability of attributes. For example, the update of a property can be represented by the following statement,

Attribute_Update_Expression:: = Update_Mode_Expression(||Attribute||)

Update_Mode_Expression::{Specified by Application Organization}

In addition, the attributes of subject, object, and environment can be dynamically modified according to specific needs and policy changes. The modification can be done in the policy creation stage, policy index stage, policy decision-making stage, or policy execution stage [24].

Several administrators in the park may formulate different policies for the access of the same user to a certain area, which may lead to restriction or conflict among multiple policies. If so, it may even cause the current policies to be unable to make normal decisions or to implement. Therefore, we should detect the existence of policy conflict first and then use the corresponding methods to resolve any policy conflict, which is the focus of the paper. It is worth noting that the premise of the research on policy conflict is that multiple permission policies should act on the same object [25,26].

Suppose that there are a set of *m* permission policies ***P*_1_**, ***P*_2_**, …, ***P_m_*** for users to access a certain area, which can be expressed as ***P*** = {***P_i_***|1 ≤ *i* ≤ *m*}. Among them, a permission policy contains *n* permission elements, which can be expressed as ***P_i_*** = {*ur_ij_*|1≤ *i* ≤ *m*,1≤ *j* ≤ *n*}. One of the permission elements is the minimum set of policies that can be enforced. For the same type of attribute and category, there may be multiple conditional constraints, and the concepts of comparable constraint items and the number of comparable constraint items need to be defined.

**Definition** **1.**
*For two different permission elements, ur_11_ and ur_21_, there is a relationship between them: equal, unequal, or inclusion, which can be expressed as ur_11_ = ur_21_, ur_11_≠ur_21_, or IsPartof(ur_11_, ur_21_), IsPartof(…, …) is the right inclusion relationship comparison function. For example, ur_11_ and ur_21_ are given to different people, if*

*ur_11_ =*
*“teacher is allowed to enter Room 101”*

*ur_21_ =*
*“teacher is allowed to enter Room 102”*

*then ur_11_ ≠ ur_21_, if*

*ur_21_ =*
*“teacher is allowed to enter the first floor”*

*then the relationship between ur_11_ and ur_21_ is IsPartof(ur_11_, ur_21_).*



**Definition** **2.**
*A comparable constraint item refers to a constraint item con that acts on the same type of attribute and belongs to the same type. Attribute i satisfies i*
*∈{s, o, e}, and category j satisfies 1 ≤ j ≤ n. s, o, and e are subject attribute, object attribute, and environment attribute, respectively. When con_i1_.typ e = con_i2_.type, con_i1_ and con_i2_ are comparable; otherwise, con_i1_ and con_i2_ are not comparable. For example,*

*con_i1_ = s.age is bigger than 18*

*con_i2_ = s.age is bigger than 16*

*then con_i1_.type = con_i2_.type = s.age, con_i1_ and con_i2_ are comparable.*



**Definition** **3.**
*For attribute i, the intersection of comparable constraints is expressed as con_i1_∩con_i2_ ≠*
*Ø. There is an intersection between two comparable constraints con_i1_ and con_i2_. The mutual exclusion of comparable constraint terms is expressed as con_i1_∩con_i2_ =*
*Ø. There is no intersection between two comparable constraint terms con_i1_ and con_i2_. In the above example,*

*con_i1_∩con_i2_ = s.age is bigger than 18 ≠*
*Ø*

*so there is an intersection between two comparable constraints con_i1_ and con_i2_.*



Merkle tree technology is used for distributed and secure storage of user operation information in smart parks, supporting real-time recording and subsequent evidence collection of data use process. In the case of epidemic prevention and control and disaster rescue, it can provide tamper-proof personnel track information and effectively track the location information of users and rescued personnel. Once a user completes an operation in the smart park, the sensor networks will generate or update the Merkle tree of the user’s operation information, and the Merkle tree of all user operation information will be saved in the sensor networks. The user operation information includes the operation user, operation time, operation type, operation data, etc. The sensor networks need to quantify this operation information. In the construction process of the Merkle tree, the user operation information is expressed in the form of the tree structure, including generating primary nodes, secondary nodes, tertiary nodes, etc., and finally generating root nodes. The mathematical transformation relationship between nodes and user behavior can be defined by themselves. Due to the correlation of nodes in the Merkle tree, the value of leaf nodes in the tree changes, which will change the value of the root node of the tree. When verifying whether the leaf node has changed, you only need to compare the correctness of the root node value. The security of the Merkle tree itself mainly depends on the security of the hash function, and the security properties such as one-way irreversibility and non-collision of hash function have long been proved, which ensures that the application of Merkle tree based on hash function is much secure and practical [27]. An enclave container can be created in the software guard extensions (SGX), and confidentiality and integrity of the code and data are secured against malicious software. A CPU in the trusted execution environment SGX can create multiple secure enclave containers for concurrent execution [28]. In the process of data flow, the generation and maintenance of Merkle run in a trusted execution environment to ensure the authenticity of the Merkle tree. Therefore, generating the log information of personnel movements based on the Merkle tree can realize efficient traceability, and the traceability results are credible and tamper-proof [29,30].

## 4. The Fine-Grained Access Control Mechanism

### 4.1. License Negotiation Interaction Protocol

The main notations used in this section and their definitions are shown in Table 1. The client agent is installed on the intelligent gateway. The sensor devices in the local area, such as cameras and RFIDs, will gather the collected user information in the intelligent gateway. The client agent on the intelligent gateway will start the corresponding license request steps based on the user’s information, to establish a license negotiation interaction process with the server PEP. The specific interaction process is shown in Figure 2. Finally, the client agent will send the opening or closing instructions to the corresponding sensor devices, such as electronic entrance guards or gates, through the results of whether it has obtained the license.

The license includes metadata, policy, key information (Keyinfo), signature, etc. The license information structure is shown in Figure 2a. Metadata information is mainly used to describe the basic information of the license. Policy is used to define a series of access policies, which are mainly expressed by authorization and condition policy elements. Keyinfo is used to store the decryption key of relevant resource content, which is generally encrypted by the public key of the license requestor to ensure the security of the key information. Signature is used to ensure the integrity and identifiability of the license, which is realized by abstract technology and digital signature technology. The entities involved in the license negotiation interaction protocol include intelligent gateway and PEP, as shown in Figure 2b.

(1)The intelligent gateway generates a license challenge message *Lic_Challenge* through the client *Agent_C_* installed on it to apply for a license from PEP.

First of all, the intelligent gateway generates the message text *Mesg* of *Lic*_*Challenge*, *Mesg* = *GenerateRequestChallenge*( ), and generates the temporary key *ck*_0_ for encrypting *Mesg*, encrypts *Mesg* with *ck*_0_ to get $Enc_{ck_{0}}(Mesg)$, and encrypts *ck*_0_ with PEP’s public key *pk_PEP_*, and signs the hash value of *Mesg* with the client’s private key *sk_C_*. The client’s certificate *Cert_C_*, the encrypted message text $Enc_{ck_{0}}(Mesg)$, the encrypted content key $Enc_{pk}{}_{{}_{PEP}}(ck_{0})$, and the signed message text hash value $Sig_{sk_{C}}(H(Mesg))$ are combined to generate *Lic_Challenge* to be transferred to PEP.
(1)$$Lic\_Challenge=\{Cert_{C}||Enc_{ck_{0}}(Mesg)||Enc_{pk}{}_{{}_{PEP}}(ck_{0})||Sig_{sk_{C}}(H(Mesg))\}$$

(2)PEP verifies the validity of the license challenge message *Lic_Challenge*, and sends the license response message *Lic_Response* to *AgentC*.

PEP decrypts $Enc_{pk}{}_{{}_{PEP}}(ck_{0})$ with the private key *sk_PEP_*, $ck_{0}’=Dec_{sk_{PEP}}(Enc_{pk}{}_{{}_{PEP}}(ck_{0}))$, decrypts $Enc_{ck_{0}}(Mesg)$ with the obtained *ck*_0′_, $Mesg’=Dec_{ck_{0}’}(Enc_{ck_{0}}(Mesg))$, and verifies whether *Mesg* is tampered or attacked by using the public key *pk_C_* of the client, i.e., $V_{1}=Ver_{pk_{C}}(H(Mesg’),Sig_{sk_{C}}(H(Mesg)))$.

If *V*_1_ passes the verification, PEP generates random number *randnum*, encrypts *randnum* with $Enc_{pk_{C}}(randnum)$, PEP’s private key *sk_PEP_* is used to sign the hash value of *randnum* to get $Sig_{sk_{PEP}}(H(randnum))$, the value of response tag is set to 1, and *Tag*, $Enc_{pk_{C}}(randnum)$ and $Sig_{sk_{PEP}}(H(randnum))$ are combined to generate license response message *Lic_Responce*.
(2)Lic_Response={Tag||EncpkC(randnum)||SigskPEP(H(randnum))}

If *V*_1_ verification fails, the value of *Tag* is set to 0, and *Tag* and error message *Error_message* are combined to generate *Lic_Response*, *Lic_Responce* = {*Tag*||*Error_message*}.

(3)The intelligent gateway *Agent_C_* verifies *Lic_Responce* returned from PEP. If the verification is passed, the license credential *Lic_Credential* is created, and then sent to PEP. The specific process is shown as follows.

The intelligent gateway uses *sk_C_* to decrypt $Enc_{pk_{C}}(randnum)$ to get *rdn*’, and verifies the integrity of *randnum* with *pk_PEP_* to get $V_{2}=Ver_{pk_{PEP}}(H(rdn’),Sig_{sk_{PEP}}(H(randnum)))$.

If *V*_2_ passes the verification, the intelligent gateway uses *sk_C_* to sign *rdn*’ to get $Sig_{sk_{C}}(rdn’)$, and encrypts the client information *Client_Info*, key ID *k_ID_*, required rights *Rights,* and signed random number $Sig_{sk_{C}}(rdn’)$. The obtained ciphertext is recorded as *Seg*1.
(3)$$Seg1=Enc_{pk_{PEP}}(Client\_Info||k_{ID}||Rights||Sig_{sk_{C}}(rdn’))$$

The intelligent gateway uses *sk_C_* to sign the hash value of *Client_Info*||*k_ID_*||*Rights*||$Sig_{sk_{C}}(rdn’)$ to get *Seg*2.
(4)$$Seg2=Enc_{sk_{C}}(H(Client\_Info||k_{ID}||Rights||Sig_{sk_{C}}(rdn’)))$$

The intelligent gateway combines *Seg*1, *Seg*2 and the user’s certificate *Cert_u_* to generate *Lic_Credential*, i.e., *Lic_Credential* = {*Cert_u_*||*Seg*1||*Seg*2}.

If *V*_2_ verification fails, the license negotiation interaction protocol stops running.

(4)PEP verifies the integrity and validity of *Lic_Credential* from the intelligent gateway. If it is valid, a license *Lic_u_* is generated and issued to the intelligent gateway through sensor networks. If it is invalid, a rejection message is returned. The specific process is as follows.

PEP decrypts the received *Seg*1 with *sk_PEP_*, *Req*_*Seg*1′$=Dec_{sk_{PEP}}(Seg1)$, uses *pk_C_* to verify the integrity of *random*, $V_{3}=Ver_{pk_{C}}(randnum,Sigsk_{C}(rdn’))$. If *V*_3_ passes the verification, PEP needs to check whether *Rights* are legal and compare with the information in the PIP, *V*_4_ = *CheckGrantPolicy*(*Lic_Credential*). If *V*_4_ passes the verification, an empty license object is generated and metadata information such as *Lic_ID_* is initialized. Generate the decryption key *key* by using the *k_ID_* and *k_Seed_* in the license server, *key* = *GenerateContentKey*(*k_ID_*,*k_seed_*), set the key attributes of the license, namely *SetAttributes*(*Lic*_0_)←{*Cert_u_*,*Rights*,*key*}, set the other attributes of the license, *SetOtherAttributes*(*Lic*_0_), encrypt the license as $Enc_{pk_{C}}(Lic_{0})$, calculate the hash value of the license and sign with *sk_PEP_* to get $Sig_{sk_{PEP}}(H(Lic_{0}))$. The combination of encrypted *Lic*_0_ and its hash signature is *Lic_u_* to be issued at present.
(5)$$Lic_{u}=Enc_{pk_{C}}(Lic_{0})||Sig_{sk_{PEP}}(H(Lic_{0}))$$

### 4.2. Protocol Security Analysis

The security of license negotiation interaction protocol is reflected in two aspects, namely, the security of license file and the security of the protocol. The security objectives of the license itself mainly include the confidentiality, tamper-proof, and unforgeability of the content. The security objectives of the interaction protocol mainly include message confidentiality, resistance to counterfeiting attacks, and non-repudiation [31,32]. This section analyzes the security and availability of the license negotiation interaction protocol proposed in this paper based on the above security objectives.

*Lic*_0_ is the main body of *Lic_u_*, including license metadata information, right information (or access control policies), key information *Key*, etc. Since PEP uses *pk_C_* to encrypt *Lic*_0_, only the corresponding *sk_C_* can be used to decrypt and obtain *key* and right information, which is expressed as
(6)$$Lic_{0}’=Dec_{sk_{C}}(Enc_{pk_{C}}(Lic_{0}))$$

Therefore, only the specified user *u* can decrypt the *Lic_u_* and read the information in the license body *Lic*_0_. The confidentiality of the license content is guaranteed. Because *Lic_u_* contains a signature for the *Lic*_0_ hash value. When *Agent_C_* receives the license, it will first verify the integrity of the license, which can be expressed as
(7)$$V=Ver_{pk_{PEP}}(H(Lic_{0}’),Sig_{sk_{PEP}}(H(Lic_{0})))$$

If other malicious third parties tamper with the content of *Lic*_0_, such as modifying the right information, key information, validity period, and so on. It will cause *H*(*Lic*_0_) ≠ *H*(*Lic*_0_′), which cannot pass the verification, resulting in the unavailability of *Lic_u_*. Therefore, once the license content is encapsulated, it cannot be changed. Any illegal changes will make the license unavailable. Before the user uses the license to access entrance guards or gates, *Agent_C_* will use *pk_PEP_* to verify the validity of the signature. It can be expressed as
(8)$$VX=Ver_{pk_{PEP}}(H(Lic_{0}’),Sig_{sk}(H(Lic_{0})))$$
where, only when *SK* = *SK_PEP_*, will it pass the verification.

PEP generates *randnum* as one of the credentials for authentication of the current protocol session. When *H*(*rdn*’) = *H*(*randnum*) and *V*_2_ = *valid*, the verification passes. The license applicant, the user *u* based on the client *Agent_C_*, will first sign *rdn*’ with *sk_C_*, then encrypt with *pk_PEP_* and encapsulate it into *Lic_Credential*. PEP verifies the returned *randnum*. If the returned *rdn*’ is equal to *randnum*, and *V*_3_ = *valid*, the verification passes. In the process of message delivery, *random* is always encrypted and digitally signed by the message sender. Therefore, its security can be guaranteed to ensure that the receiver of the license is the user applying for the license *u*. In addition, in the process of protocol interaction, the receiver of the message will verify the digital signature of the message sender to authenticate the identity of the message sender. Therefore, in the process of license negotiation interaction, malicious third parties cannot impersonate the participants of the agreement.

The user *u* cannot deny the license challenge messages *Lic_Challenge* and the license credential message *Lic_Credential* sent by itself to PEP. When applying for a license, user *u* first transmits *Cert_C_* to PEP and signs *H*(*Mesg*) with *sk_C_*, so that PEP can determine the reliability of the source client of the license challenge message. When the user *u* gets the response from the PEP, he transmits *Cert_u_* to the PEP, and uses *sk_PEP_* to sign the hash value *H*(*Credential_Text*) so that PEP can determine the reliability of the source user of the license credential message. Therefore, with *Lic_Challenge* and *Lic_Credential*, we can identify the message as being sent from *Agent_C_* by user *u*, and user *u* cannot deny it. In addition, PEP cannot deny that it has issued licenses to users *u*. The certification process is like the above process, so it will not be repeated.

In conclusion, during the license negotiation and interaction between PEP and *Agent_C_* (or user *u*), neither party can deny the messages sent to the other party by itself.

### 4.3. Policy Conflict Detection and Resolution Algorithm

Inconsistent permissions defined in different policies in PAP may cause PDP policy conflict, and its detection algorithm is shown in Algorithm 1.
**Algorithm 1.** The algorithm of permission availability detection.  Input: Permission element *ur_ij_*, 1 ≤ *i* ≤ *m*, 1 ≤ *j* ≤ *n*  Output: Permissions available or unavailable     Result = Available; // The result is initialized to available rights.      for *i* = 1:*m*, do         for *x* = *i* + 1:*m* do          for *j* = 1:*n* do           for *y* = *j* + 1:*n* do            if *ur_ij_* = *ur_xy_* or IsPartof(*ur_ij_*, *ur_xy_*) then            // The two permission elements are equal or contain              result = Available;             return result;            else if (*ur_ij_* ≠ *ur_xy_*)then             result = NotAvailable;           end if           end for         end for        end for     end for     return result

The mutual exclusion between the condition constraints of multiple policies for the same comparable constraint term will make the policy unable to be correctly decided or executed. The detection algorithm is shown in Algorithm 2.
**Algorithm 2.** The algorithm of conditional compatibility detection.  Input: Comparable constraint item *con_ij_*, *i* ∈ {*s*, *o*, *e*}, *j* ∈ {*j*|1 ≤ *j* ≤ *n*}  Output: Conditional intersection (false) or exclusion (true)     result = false; // The result is initialized to conditional intersection     for each *i* ∈ {*s*, *o*, *e*}, do         for *j* = 1:*n* do          for *k* = *j* + 1:*n* do              if *con_ij_*.type = *con_ik_*.type then               if *con_ij_*∩*con_ik_* ≠ Ø then               result=true;    //  Conditional exclusion              end if             end if         end for        end for       end for     end for     return result

Policy conflict resolution can be carried out either in the policy creation stage or in the policy decision-making stage, which can effectively avoid conflicts between policies.

### 4.4. Tamper-Proof Access Record Mechanism

Based on the trusted execution environment, the paper designs a tamper-proof access record mechanism based on the Merkle tree. In distributed storage, the stored data will have multiple copies and be distributed to different storage devices. In order to maintain data consistency, replica synchronization is required, and the first thing is to compare those current replicas and find out whether they are consistent or not [33]. If inconsistent, find out the inconsistencies and synchronize them. Meanwhile, if the root hash of the Merkle tree is consistent, the data is the same. If the root hash is inconsistent, the inconsistent data can be quickly retrieved through the Merkle tree.

The personnel or equipment movement trajectory traceability protocol based on the Merkle tree is shown in Figure 3. For the moving tracks of personnel and equipment, sensor networks generate tamper-proof and real-time log records based on the Merkle tree, which can support the forensics of the moving tracks of personnel and equipment in the smart park afterward. By designing a movement trajectory traceability protocol based on the Merkle tree, online verification log records and offline storage of encrypted information are realized. The user identity is identified by adding the user certificate, and the shared key between the user and SGX is calculated by using the elliptic curve Diffie Hellman key exchange (ECDH) to realize the data sharing participated by multiple users [34].

The protocol includes three parts, i.e., data encryption and upload, data decryption, and log inspection.

The intelligent gateway, *Agent_C_*, uses the key to encrypt the data and upload it to the storage device, in which the decryption key is only calculated by the decryption device enclaves. The storage device will store the encrypted data indexed by its hash, that is, each log information *r_i_* can be retrieved through hash *Hash*(*r_i_*). After uploading data, *Agent_C_* will keep all its ciphertext records ***R***
*=* {*r*_0_, *…*,*r_n_*}, or only its ciphertext hash value ***R****’ =* {*Hash*(*r_0_*), *...*,*Hash*(*r_n_*)}.

Data decryption is the main part of tamper-proof access record mechanism. The data requester starts the decryption protocol by sending a request for the data ciphertext hash identifier request *request*(*Hash*(*r_i_*)*,*
*Cert_u_*) to the storage device. *r_i_* represents the requested log information, and *Cert_u_* the certificate of the data requester. The storage device forwards the request *log*(*Hash*(*r_i_*), *Cert_u_*) to the log server. The log server needs to store at least the hash identifier of the requested data. In addition, it can also store other more information, such as the identity of the data requester, the time of requesting data, etc. The log server adds this new request into the Merkle tree. At the same time, the log server needs to generate two proofs. The existence proof π ensures that the new request is indeed included in the new tree. The extended proof ρ ensures that the new tree *H’* is indeed an extension of the old tree *H*. The log server stores these five elements (*H*, *H*’, *π*, *ρ*, *Cert_u_*) back to the storage device. The storage device forwards *r_i_* and the above five elements to the decryption device. The decryption device needs to check whether the current decrypted data is consistent with the version in the log record according to these data. After receiving (*r_i_*, *H*, *H*’, *π*, *ρ*, *Cert_u_*), the enclaves in the decryption device first load the *H* corresponding to the user *u* from the storage device into the enclaves according to the incoming *Cert_u_*, and then verify the proof *π* and *ρ*. If the verification passes, the loaded value *H* is updated to *H*’, enclaves store *H*’ in the storage device, and calculates the decryption key to perform decryption. If the verification fails, the protocol is stopped. The decryption device calculates the decryption key corresponding to the requested data, performs the decryption operation *dec*(*r_i_*) *= data*(*r_i_*), and sends back the decrypted result to user *u*.

The user needs to check the log record to get all the information about data decryption. The requirements of this step go as follows. At first, the user calculates the root value of the current Merkle tree, compares it with the log record, and checks the correctness of the log record. Once the decryption device completes this decryption, it updates the Merkle tree root *H*, signs with the signature key to obtain *Sign*(*H*), and forwards it to the storage device. Finally, the storage device forwards the decryption request *Hash*(*r_i_*) and the signed Merkle tree root *Sign*(*H*) to user *i*. Therefore, users can trace and be accountable for the movement trajectory of users and equipment.

## 5. Performance Evaluation

### 5.1. Experimental Setup

In order to evaluate the method proposed in this paper, we do the experiment by adopting the face recognition interface of Baidu. Face recognition is not the focus of this research, so the existing technologies are used for the experiment. We use the devices including one Lenovo server SR550, one Raspberry PI edge computing box, five Hikvision cameras, five intelligent door locks, two desktop computers and one laptop to build the security control model to simulate the real environment of smart park. We also use IntelliJ IDEA 2021 and Eclipse for programming, and the programming language is Java.

In the initialization phase, the camera is used to collect user’s facial information, and Baidu application programming interface (API) is invoked to upload pictures. Then corresponding permission levels are configured for users, intelligent door locks, and accessible computer resources, and permission control policies are set. When a user submits an access request, the API will be used for identity authentication. If the user passes the authentication, the access request will respond according to the permission control policy.

The public data set of face recognition based on machine learning is as shown in Table 2. The parameters that need to be optimized are the hyperparameter (alpha value), and hyperparameters are the parameters used to control the behavior of the algorithm while building the model. The methods commonly used to optimize hyperparameters including traditional manual tuning, grid search, and random search. In traditional manual tuning, we manually check with random sets of hyper-parameters by training the algorithm and select the best set of parameters that fits our objective. The grid search is a basic hyper-parameter tuning technique. It is similar to manual tuning, where it builds a model for each permutation of all the given hyperparameter values specified in the grid, evaluates, and selects the best model. The motivation to use random search in place of grid search is that for many cases, all the hyperparameters may not be equally important. Random search selects a random combination of parameters from the hyper-parameter space. Therefore, random search has empirically been demonstrated to give better results than traditional manual tuning and grid search.

### 5.2. Experimental Result

Table 3 shows the functional comparison between the access control mechanism proposed in this paper and other literature [35,36]. The Jada mechanism proposed in this paper supports multi-party authorization and traceability at the same time. IBAC and ciphertext policy access control (CPAC) are usually in charge of a single authority, and IBAC can no longer meet the dual requirements of the access control system for function and performance. Multi-authority access control (MAAC) can coordinate different access control policies among multiple parties, realize unified access control, and support the fusion and conflict detection of different access control policies [37]. In [35], algebraic operators are constructed, including addition, intersection, and subtraction, to abstract the synthesis of different policies, to realize the formal description of access control policy synthesis. However, simple set intersection, union, and difference operations cannot accurately reflect the real synthesis strategy and do not have traceability. In the MAAC mechanism, user permissions are only related to their attributes, which can hide the user’s real identity information and provide effective and reliable access control. However, because of this strong anonymity, it is convenient for malicious users to abuse their rights. Users with different attributes can obtain new attribute sets through collusion to obtain the permissions corresponding to the new attributes. Malicious users can distribute their own attribute private keys to users whose other attributes are not satisfied. The third party of attribute management can generate permissions to access all resources according to the attributes required by the access policy. When these malicious events occur, they are generally unable to detect these behaviors, let alone accurately associate malicious users [38]. Jada proposed in the paper has traceability and better fills the shortage of MAAC.

In the sensor networks environment, the time overhead caused by adopting different security control schemes is shown in Figure 4. It mainly counts the times of exponential operation and bilinear pairing used in initialization, encryption, key generation, and decryption and traceability protocol. Compared with the MAAC, in order to achieve traceability, the overhead of the scheme in initialization, encryption, and key generation steps is basically the same as that in [35]. There is only one additional exponential operation overhead in initialization and encryption steps, and the key generation steps are basically the same. According to MAAC, the decryption overhead of the user is constant. In the next step, this paper will also consider how to reduce the decryption overhead of the user on the premise of supporting traceability. Both CPAC and JADA support traceability. The difference is that CPAC only supports a single authorized institution. The increased overhead in this paper is mainly related to the authorization authority in the system and has nothing to do with the number of attributes involved in each step. To sum up, the overhead of this scheme is acceptable. In this paper, an elliptic curve group is selected as the operation basis of Jada, and its group order is the product of three 517-bit primes [39]. The experimental simulation results verify the consistency between the performance of the scheme and the theoretical analysis.

The finer the security control granularity of the smart park is, the higher the complexity of the permission policy. The complexity of the permission policy is related to the permission elements contained in a policy and the number of comparable constraints in conditional expressions. In the experimental test, 100 policies were written according to the access control requirements of the smart park for users, in which each policy contains 1–5 permission elements and 1–5 comparable constraints. These policies are implemented according to IBAC, MAAC, CPAC, and JADA mechanisms, respectively. We verify the accuracy, fine granularity, and efficiency of security control of these policies. The decision time required to centrally process these strategies are recorded in the PDP. The experimental results of policy execution accuracy and policy decision time are shown in Figure 5a,b, respectively. As can be seen from the Figure, IBAC cannot support complex policies, although it is superior to the other three mechanisms in policy execution accuracy and decision-making time. Compared with CPAC and MAAC, the average strategy execution accuracy of JADA is about 94%, and the decision-making time is shortened by 14.6 and 23.8% when 50 policies are processed.

## 6. Conclusions

The smart park refers to a modern park that makes full use of sensor networks technology to provide intelligent social management and services. At present, owners, tenants, visitors, property management, and other personnel flow normally in the park. Maintenance, express delivery, takeout, cleaning, and other services cause complex identities and frequent cross-domain activities. It is difficult for the park management side to restrict the space, time, and frequency of personnel activities. In the case of public security, epidemic, and other factors, it is impossible to quickly trace the source of personnel activities, and these factors will lead to new safety problems. So there exist potential safety hazards in park governance.

The paper designs a security control method based on cryptography to adapt to the characteristics of dynamic personnel movement, which can solve problems such as the difficulty of a fine description of personnel permissions and the conflict of multi-party joint authorization strategies on the premise of ensuring the privacy information of personnel in the park, so that the owner can accurately specify the allowable activity scope for foreign users, avoid the serious phenomenon of over authorization and insufficient authorization, and realize the principle of minimizing authorization for foreign users. In case of public security incidents, it can efficiently trace the source and trace the responsibility of the park personnel. By introducing a tamper-proof recording mechanism, it can solve the problems of malicious attacks by internal personnel, the bottleneck of trusted third-party dependence, and the high overhead of tracking algorithm. The method proposed in the paper realizes the fine authority control before the personnel flow in the park and the whole process clarification of the activity track after the flow to ensure the orderly and free flow of personnel according to regulations.

This paper mainly studies authorization, access control, and movement trajectory traceability. Our mechanism supports multi-party authorization and has traceability, so the time overhead has increased a little compared with IBAC and MAAC, which can be optimized in the future. For further research, we would like to apply multi-factor authentication technology in identity authentication, such as RFID and face recognition, and in the face recognition module, we can increase the function of classifying gender and age. Meanwhile, private information, such as personal identity information and movement trajectory needs to be protected, which can be realized by homomorphic encryption, oblivious transfer, and so on.

## Figures and Tables

**Figure 1 sensors-21-06815-f001:**
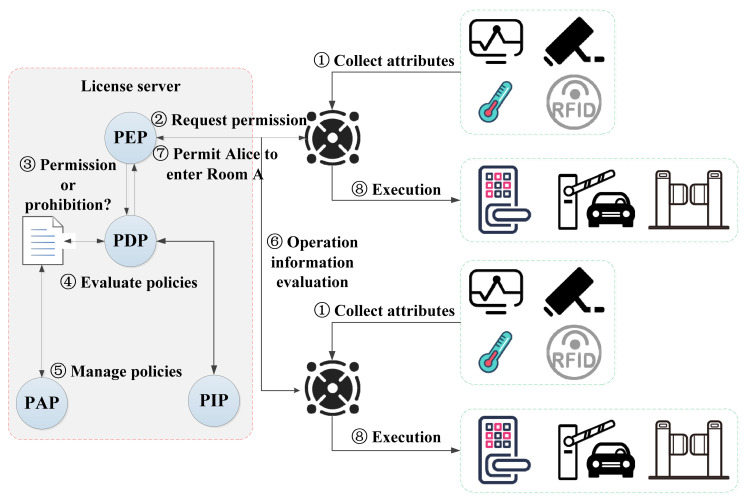
A sensor networks-based security control model of smart parks.

**Figure 2 sensors-21-06815-f002:**
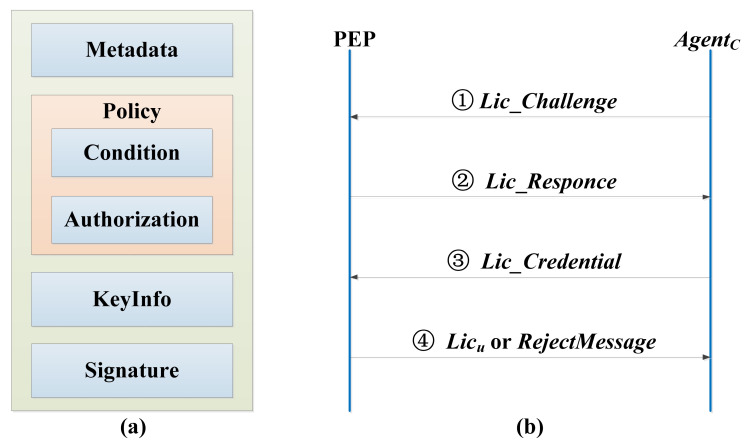
The license information structure and license negotiation interaction protocol. (**a**) License information structure; (**b**) license negotiation interaction protocol.

**Figure 3 sensors-21-06815-f003:**
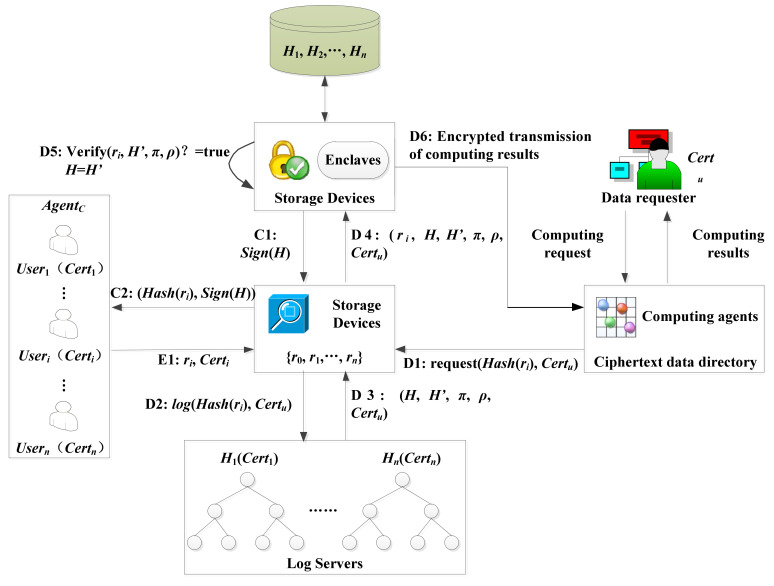
The movement trajectory traceability protocol based on a Merkle tree.

**Figure 4 sensors-21-06815-f004:**
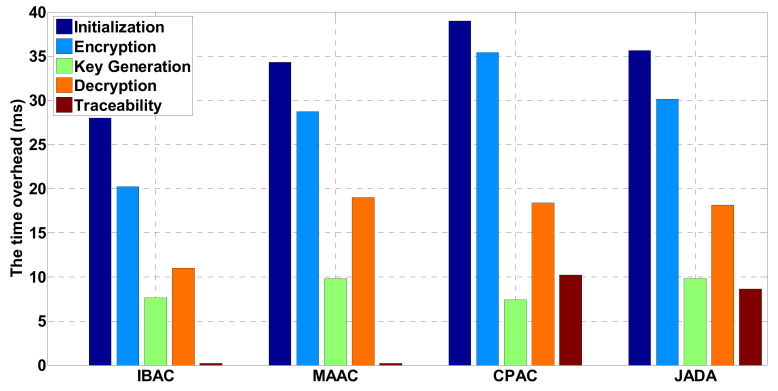
The time overhead of different security management and control schemes in sensor networks.

**Figure 5 sensors-21-06815-f005:**
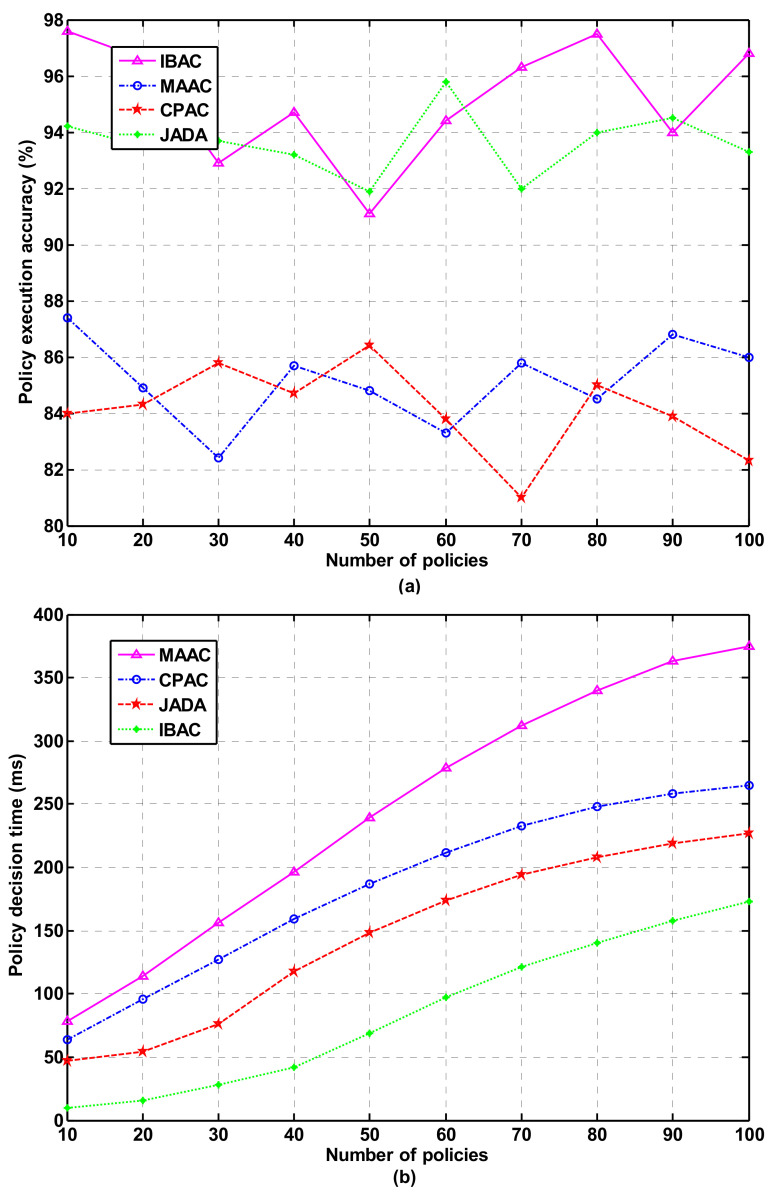
Policy execution accuracy and policy decision time in sensor networks. (**a**) Policy execution accuracy; (**b**) policy decision time.

**Table 1 sensors-21-06815-t001:** Main notations.

Notation	Definition
*ck* _0_	Temporary key generated by the gateway
*pk_PEP_*	PEP’s public key
*sk_PEP_*	PEP’s private key
*pk_C_*	Client’s public key
*sk_C_*	Client’s private key
*Cert_C_*	Client’s certificate
*k_ID_*, *k_Seed_*	Key ID, the seed used to generate the key
*Lic* _0_	License generated by PEP
*Enc_key_(message)*	Encrypt message with key
*H(message)*	Hash value of message
*Sig_key_(message)*	Sign message with key
*Dec_key_(message)*	Decrypt message with key

**Table 2 sensors-21-06815-t002:** Public large-scale face datasets.

Dataset	Images	Website Address
CelebFaces+	202,599	https://mmlab.ie.cuhk.edu.hk/projects/CelebA.html (Accessed on 30 March 2021)
UMDFaces	367,920	https://www.umdfaces.io/ (Accessed on 30 March 2021)
CASIA-WebFace	494,414	https://www.cbsr.ia.ac.cn/english/CASIA-WebFace-Database.html (Accessed on 30 March 2021)
VGGFace	2.6 mil.	https://www.robots.ox.ac.uk/~vgg/data/vgg_face/ (Accessed on 30 March 2021)
VGGFace2	3.3 mil.	https://www.robots.ox.ac.uk/~vgg/data/vgg_face2/ (Accessed on 30 March 2021)
MegaFace	4.7 mil.	https://megaface.cs.washington.edu/dataset/download.html (Accessed on 30 March 2021)
MS-Celeb-1M	10 mil.	https://msceleb.org/(Accessed on 30 March 2021)

**Table 3 sensors-21-06815-t003:** Comparison of relevant access control mechanisms.

References	Mechanisms	Multi-party Authorization	Traceability
[4]	IBAC	No	No
[35]	MAAC	Yes	No
[36]	CPAC	No	Yes
Our paper	JADA	Yes	Yes

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
