# Peer review of "A Security Management and Control Solution of Smart Park Based on Sensor Networks"

_sensors, 2021, doi:10.3390/s21206815_

Round 1

Reviewer 1 Report

The main goal of research is to resolving internal personnel malicious attack using  Merkle tree-based movement trajectory traceability protocol. Overall, this manuscript reads well and has the potential to provide us some useful implications for future direction. However, there are some important issues that the authors have to address in order to make this manuscript better.

My major comments to this manuscript are as follows:

  1. The logic of introduction is OK. It is good to state the contribution of this study in the introduction.
  2. Good research needs to explain what it means in practice, so the managerial implications of this study should be stressed in this study.
  3. The authors must separate related work from Introduction. The authors have not mentioned their real drawbacks or the need of the proposed model. This section must include recent works.
  4. Please clarify definitions 1, 2 and 3 with examples.
  5. Please add a table that includes all used variables and notations.
  6. The author has to include the details about the Implementation environment and tool details used for the proposed work.
  7. In section 4, Performance Evaluation. The experiment process should be descripted clearly.
  8. The limitation and future research direction should be added at the end of this paper. The limitation can include the limitations of security control and the methodology, etc. The future research direction can involve the next steps to deepen this topic.

Reviewer 2 Report

Authors of the work “A Security Management and Control Solution of Smart Park Based on Sensor Networks” present in this paper a joint authorization and dynamic access control mechanism, providing unified identity management services, access control services and policy management services, and effectively solving the problem of multi-authorization in user identity and authority management.

Overall, it is a well-structured paper; the introduction section is wide, and presents the purpose of the research in detail. There is a comparison and evaluations of the proposed method, using proper figures and showing up the proofs of the experiments included in the paper.

Although the proposal is interesting and within the scope of Sensors Journal, there are different issues that should be addressed in order to improve the work.

[Minor Comments]

  • The state of the art should be more extensive, as it only shows a few examples of how this problem has been tackled previously. A larger number of works should be presented with their advantages and shortcomings, results achieved, etc.
  • The functionality of the agents is not clear, can you provide more information about their functionality, are the well-known agent systems used as multi-agent systems? Or is it another agent concept? Please elaborate on this aspect.
  • A clearer comparison is required (cross-validation, etc, …). It should be appreciate the inclusion of statistical significance tests in order to compare the Mechanisms, you still need to report (i) sample size, (ii) alpha value, and clearly state the null and alternative hypotheses. In order to improve the comparative, It would be interesting to look how this is done in the following paper:
    • González-Briones, A., Villarrubia, G., De Paz, J. F., & Corchado, J. M. (2018). A multi-agent system for the classification of gender and age from images. Computer Vision and Image Understanding.
    •  
  • The article's English must be reviewed by a native speaker.

Reviewer 3 Report

Construction of the phrase with ‘’… lack of audit evidence collection and so on. "It is not very suitable for a scientific paper after my opinion in special in abstract construction.

References to the bibliography "[5] introduces the current authentication .." in the start of the sentence must be changed to the names of the authors or the name of the publication.

The conclusions must also provide a comparison of the results obtained with the initial ones.

The results must be highlighted because the proposed method is not observed compared to other existing ones and the improvements it brings.

Reviewer 4 Report

Dear authors,

It is an interesting work on a very relevant topic. However, I would like to draw attention to some points below:

In sections 2 and 3, it is possible to observe the authors' concern to present the System Model and The Fine-grained Access Control Mechanism in detail. However, in section 4, "Performance Evaluation," more detailed information needs to be presented.

  1. What method was used to test performance?
  2. The authors state:
  3. a) "this paper proposes a joint authorization and dynamic access control mechanism (JADA) for the scenario in which people and devices move in different areas of smart parks." Has the system been tested for various possibilities of moving people and devices? What was the strategy used to ensure that the necessary scenarios were considered? The reader will need more information to be sure of the robustness of the set of scenarios considered.
  4. b) "The license negotiation interaction protocol is designed to prevent the common network attacks on the process of identity authentication and authority management." How was this tested?
  5. What are the limitations of your research? How did they impact the results?
  6. The text of the Author Contributions, Data Availability Statement, and Acknowledgments fields have not changed (the template text remains).

Round 2

Reviewer 1 Report

The authors have satisfactorily responded to all my questions and my decision is accept.

Reviewer 4 Report

Reviewer Comments to Author

Dear authors,

Thank you for addressing my comments. The manuscript is significantly improved.

With kind regards,

Carlos Soares